

# Collecting reliable clades using the Greedy Strict Consensus Merger

Markus Fleischauer and  Sebastian Böcker

Lehrstuhl für Bioinformatik, Friedrich-Schiller Universität, Jena, Thüringen, Germany

## ABSTRACT

Supertree methods combine a set of phylogenetic trees into a single supertree. Similar to supermatrix methods, these methods provide a way to reconstruct larger parts of the Tree of Life, potentially evading the computational complexity of phylogenetic inference methods such as maximum likelihood. The supertree problem can be formalized in different ways, to cope with contradictory information in the input. Many supertree methods have been developed. Some of them solve NP-hard optimization problems like the well-known Matrix Representation with Parsimony, while others have polynomial worst-case running time but work in a greedy fashion (FlipCut). Both can profit from a set of clades that are already known to be part of the supertree. The Superfine approach shows how the Greedy Strict Consensus Merger (GSCM) can be used as preprocessing to find these clades. We introduce different scoring functions for the GSCM, a randomization, as well as a combination thereof to improve the GSCM to find more clades. This helps, in turn, to improve the resolution of the GSCM supertree. We find this modifications to increase the number of true positive clades by 18% compared to the currently used Overlap scoring.

# INTRODUCTION

Supertree methods are used to combine a set of phylogenetic trees with non-identical but overlapping taxon sets, into a larger supertree that contains all the taxa of every input tree. Many supertree methods have been established over the years, see for example: *Bininda-Emonds* (*2004*); *Ross & Rodrigo* (*2004*); *Chen et al.* (*2006*); *Holland et al.* (*2007*); *Scornavacca et al.* (*2008*); *Ranwez, Criscuolo & Douzery* (*2010*); *Bansal et al.* (*2010*); *Snir & Rao* (*2010*); *Swenson et al.* (*2012*); *Brinkmeyer, Griebel & Böcker* (*2013*); *Berry, Bininda-Emonds & Semple* (*2013*); *Gysel, Gusfield & Stevens* (*2013*); *Whidden, Zeh & Beiko* (*2014*); these methods complement supermatrix methods which combine the "raw" sequence data rather than the trees (*Von Haeseler*, *2012*).

In contrast to supermatrix methods, supertree methods allow us to analyze large datasets without constructing a multiple sequence alignment for the complete dataset, and without a phylogenetic analysis of the resulting alignment. In this context, supertree methods can be used as part of divide-and-conquer meta techniques (*Huson, Nettles & Warnow*, *1999*; *Huson, Vawter & Warnow*, *1999*; *Roshan et al.*, *2004*; *Nelesen et al.*, *2012*), which break down a large phylogenetic problem into smaller subproblems that are

Corresponding author
Markus Fleischauer,
markus.fleischauer@uni-jena.de

computationally much easier to solve. The results of the subproblems are then combined using a supertree method.

Constructing a supertree is easy if no contradictory information is encoded in the input trees (*Aho et al.*, *1981*). However, resolving conflicts in a reasonable and swift way remains difficult. Matrix Representation with Parsimony (MRP) (*Baum*, *1992*; *Ragan*, *1992*) is still the most widely used supertree method today, as the constructed supertrees are of comparatively high quality. Since MRP is NP-hard (*Foulds & Graham*, *1982*), heuristic search strategies have to be used. *Swenson et al.* (*2012*) introduced SuperFine which combines the Greedy Strict Consensus Merger (GSCM) (*Huson, Vawter & Warnow*, *1999*; *Roshan et al.*, *2003*) with MRP. The basic idea is to use a very conservative supertree method (in this case GSCM) as preprocessing for better-resolving supertree methods (in this case MRP). Conservative supertree methods only resolve conflict-free clades and keep the remaining parts of the tree unresolved. We call those resolved parts of a conservative supertree *reliable clades*. Other better-resolving supertree methods, such as the polynomial-time FLIPCUT (*Brinkmeyer, Griebel & Böcker*, *2013*) algorithm, may also benefit from this preprocessing.

The number of *reliable clades* returned by GSCM is highly dependent on the merging order of the source trees. Although the GSCM only returns clades that are compatible with all source trees, we find that it likewise produces clades which are not supported by any of the source trees (*bogus clades*). Obviously, bogus clades do not necessarily have to be part of the supertree.

With the objective of improving the GSCM as a preprocessing method, we introduce new scoring functions, describe a new randomized GSCM algorithm, and show how to combine multiple GSCM results. Our new scorings increase the number of true positive clades by 5% while simultaneously reducing the number of false positive clades by 2%. Combining different scoring functions and randomization further increases the number of true positive clades by up to 18%. We find that combining a sufficient number of randomized GSCM trees is more robust than a single GSCM tree.

We describe and implement a variant of the GCSM algorithm for rooted input trees and adapt the scoring functions used within SuperFine (*Swenson et al.*, *2012*). We find that our new scoring functions and modifications improve on the ones adapted from *Swenson et al.* (*2012*) in the rooted case. Although all scoring functions and modifications can be generalized to the unrooted case, the results may differ for unrooted trees.

All presented methods are part of our GSCM command line tool (https://bio.informatik. uni-jena.de/software/gscm/).

## METHODS

### Preliminaries

In this paper, we deal with graph theoretical objects called rooted (phylogenetic) trees. Let $\mathcal{V}(T)$ be the vertex set. Every leaf of a tree $T$ is uniquely labeled and called a *taxon*. Let $\mathcal{L}(T) \subset \mathcal{V}(T)$ be the set of all taxa in $T$. We call every vertex $v \in \mathcal{V}(T) \setminus \mathcal{L}(T)$ an inner vertex. An inner vertex $c \in \mathcal{V}(T)$ comprises a clade $C = \mathcal{L}(T^c) \subseteq \mathcal{L}(T)$ where $T^c$ is the

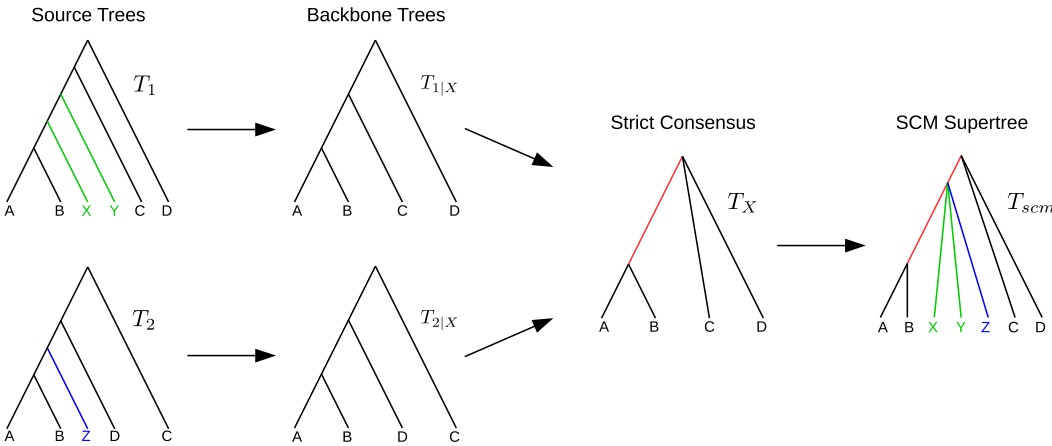

**Figure 1  Example SCM run including collision handling.** The backbone trees $T_{1|X}$ and $T_{2|X}$ are merged using the strict consensus. The remaining subtrees of $T_1$ and $T_2$ are colored in green and blue, respectively. Both subtrees attach to the same edge in $T_X$ (red). The green and blue subtrees are inserted into $T_X$ by generating a polytomy (collision handling).

subtree of $T$ rooted at $c$. Two clades $C_1$ and $C_2$ are *compatible* if $C_1 \cap C_2 \in \{C_1, C_2, \emptyset\}$. Two trees are compatible if all clades are pairwise compatible. The *resolution* of a rooted tree is defined as $\frac{|\mathcal{V}(T)| - |\mathcal{L}(T)|}{|\mathcal{L}(T)| - 1}$. Hence, a completely unresolved (i.e., star) tree has resolution 0, whereas a fully resolved (i.e., binary) tree has resolution 1. For a given collection of trees $\mathcal{T} = \{T_1, \ldots, T_k\}$, a supertree $T$ of $\mathcal{T}$ is a phylogenetic tree with leaf set $\mathcal{L}(T) = \bigcup_{T_i \in \mathcal{T}} \mathcal{L}(T_i)$. A supertree $T$ is called a *consensus tree* if for all input trees $T_i, T_j \in \mathcal{T}$, $\mathcal{L}(T_i) = \mathcal{L}(T_j)$ holds. A *strict consensus* of $\mathcal{T}$ is a tree that only contains clades present in all trees $T_i \in \mathcal{T}$. A *semi-strict consensus* of $\mathcal{T}$ contains all clades that appear in some input tree and are compatible with each clade of each $T_i \in \mathcal{T}$ (*Bryant, 2003*). For a set of taxa $X \subset \mathcal{L}(T)$, we define *the X-induced subtree of T*, $T_{|X}$ as the tree obtained by taking the (unique) minimal subgraph $T(X)$ of $T$ that connects the elements of $X$ and then suppressing all vertices with out-degree one: that is, for every inner vertex $v$ with out-degree one, replace the adjacent edges $(p, v)$ and $(v, c)$ by a single edge $(p, c)$ and delete $v$.

## Strict consensus merger (SCM)

For a given pair of trees $T_1$ and $T_2$ with overlapping taxon sets, the SCM (*Huson, Vawter & Warnow, 1999*; *Roshan et al., 2003*) calculates a supertree as follows. Let $X = \mathcal{L}(T_1) \cap \mathcal{L}(T_2)$ be the set of common taxa and $T_{1|X}$ and $T_{2|X}$ the $X$-induced subtrees. Calculate $T_X = \text{STRICTCONSENSUS}(T_{1|X}, T_{2|X})$. Insert all subtrees, removed from $T_1$ and $T_2$ to create $T_{1|X}$ and $T_{2|X}$, into $T_X$ without violating any of the clades in $T_1$ or $T_2$. If removed subtrees of $T_1$ and $T_2$ attach to the same edge $e$ in $T_X$, a collision occurs. In that case, all subtrees attaching to $e$ will be inserted at the same point by subdividing $e$ and creating a polytomy at the new vertex (see Fig. 1).

Note that neither the strict consensus nor the collision handling inserts clades into the supertree $T_X$ that conflict with any of the source trees.

## Greedy Strict Consensus Merger (GSCM)

The GSCM algorithm generalizes the SCM idea to combine a collection $\mathscr{T} = \{T_1, T_2, \ldots, T_k\}$ of input trees into a supertree $T$ with $\mathscr{L}(T) = \bigcup_{i=1}^{k} \mathscr{L}(T_i)$ by pairwise merging trees until only the supertree is left. Let $score(T_i, T_j)$ be a function returning an arbitrary score of two trees $T_i$ and $T_j$. At each step, the pair of trees that maximizes $score(T_i, T_j)$ is selected and merged, resulting in a greedy algorithm. Since the SCM does not insert clades that contradict any of the source trees, the GSCM returns a supertree that only contains clades that are compatible with all source trees.

### Algorithm 1. Strict Consensus Merger

```
1: function SCM(tree T₁, tree T₂)
2:     X ← 𝓛(T₁)∩𝓛(T₂)
3:     if |X| ≥ 3 then                          ▷ Otherwise, the merged tree will be unresolved.
4:         calculate T₁|X and T₂|X
5:         T_X ← STRICTCONSENSUS(T₁|X, T₂|X)
6:         for all removed subtrees of T₁ and T₂ do
7:             if collision then    ▷ Subtrees of T₁ and T₂ attach to the same edge e in T_X (Fig. 1)
8:                 insert all colliding subtrees at the same point on e by generating a polytomy.
9:             else
10:                Reinsert subtree into T_X without violating any of the bipartitions in T₁ or T₂.
11:            end if
12:        end for
13:        return T_X
14:    end if
15: end function
```

### Algorithm 2. Greedy Strict Consensus Merger

```
1: function PICKOPTIMALTREEPAIR(trees 𝒮 ⊆ {T₁, T₂, ..., T_k})
2:     Pick two trees {T_i, T_j} ⊆ 𝒮 which maximize score(T_i, T_j)
3:     return T_i, T_j
4: end function
1: function GSCM(trees {T₁, T₂, ..., T_k})
2:     𝒮 ← {T₁, T₂, ..., T_k}
3:     while |𝒮| ≥ 2 do
4:         T_i, T_j ← PICKOPTIMALTREEPAIR(𝒮)
5:         𝒮 ← 𝒮 \ {T_i, T_j}
6:         T_scm ← SCM(T_i, T_j)
7:         𝒮 ← 𝒮 ∪ {T_scm}
8:     end while
9:     return T_scm
10: end function
```

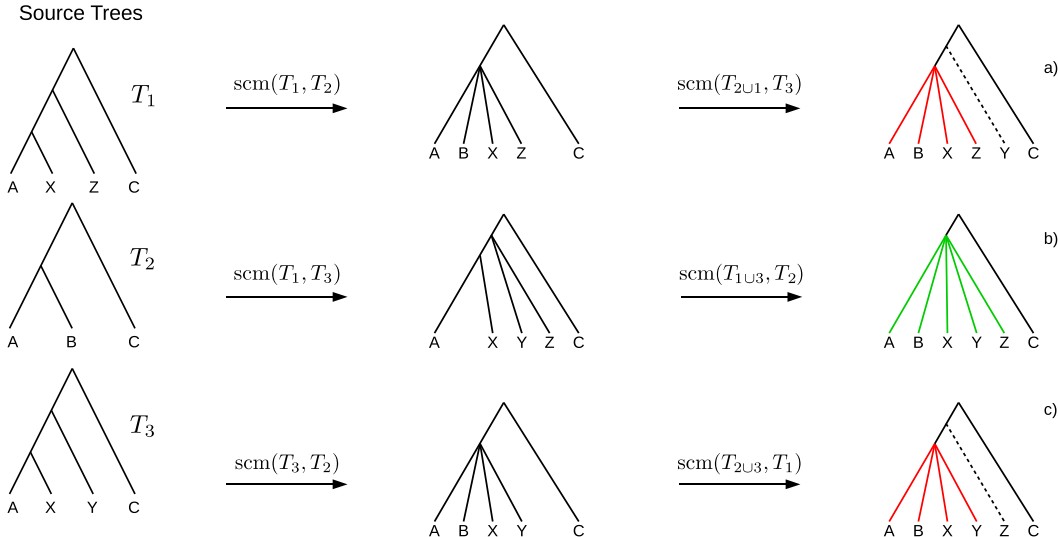

Source Trees

**Figure 2  Example where the collision handling inserts bogus clades (red) into the supertree.** Bogus clades are induced by obviated collisions, which are prevented by a previous collision on the same edge. Supertrees (A) and (C) are estimated on the same set of source trees, but contain conflicting clades ((ABXZ) conflicts with (ABXY)) induced by different merging orders. The correct supertree is (B).

## Tree merging order

Although the SCM for two trees is deterministic, the output of the GSCM is influenced by the order of selecting pairs of trees to be merged, since the resulting number and positions of collisions may vary.

Let $T_1, \ldots, T_n$ be a set of input trees we want to merge into a supertree using the GSCM. When merging two trees, the strict consensus merger (SCM) accepts only clades, that can be safely inferred from the two source trees. In case of a collision during reinsertion of unique taxa, the colliding subtrees are inserted as a polytomy on the edge where the collision occurred.

If collisions of different merging steps occur on the same edge, the polytomy created by the first collision may cause the following collisions to not occur. Such obviated collisions induce *bogus clades* (see Fig. 2) which cannot be inferred unambiguously from the source trees and hence should not be part of the supertree. A clade $C$ of a supertree $T = \mathrm{GSCM}(T_1, \ldots, T_n)$ is a *bogus clade* if there is another supertree $T' = \mathrm{GSCM}(T_1, \ldots, T_n)$ (based on a different tree merging order) that contains a clade $C'$ conflicting with $C$ (see Figs. 2A and 2C). Note that bogus clades cannot be recognized by comparison to the source trees since they do not conflict with any of the source trees $T_1, \ldots, T_n$. All clades in the GSCM supertree that are not bogus, are called *reliable clades*.

Because of these bogus clades the GSCM supertree with the highest resolution may not be the best supertree. To use the GSCM as preprocessing for other supertree methods, it is important to prevent bogus clades. Clades resulting from the preprocessing are fixed and will definitely be part of the final supertree (even if they are wrong). To use GCSM as an efficient preprocessing we want to determine a preferably large amount of the existing reliable clades. Therefore, we searched for scoring functions that maximize the number of reliable clades by simultaneously minimizing the number of bogus clades.

## Scoring functions

We present three novel scoring functions that produce high quality GSCM supertrees with respect to $F_1$-score and number of unique clades (unique in terms of not occurring in a supertree resulting from any of the other scorings). In addition, we use the original *Resolution* scoring (*Roshan et al.*, *2003*), as well as the *Unique-Taxa* and *Overlap* scorings (*Swenson et al.*, *2012*).

Let $uc(T, T') = \mathcal{V}(T_{|\mathcal{L}(T)\setminus\mathcal{L}(T')} \setminus \mathcal{L}(T))$ be the set of unique clades of $T$ compared to $T'$.

**Unique-Clades-Lost scoring:** minimizing the number of unique clades that get lost:

$$score(T_i, T_j) = -\Big(\big(|uc(T_i, T_j)| - |uc(scm(T_i, T_j), T_j)|\big)$$
$$+ \big(|uc(T_j, T_i)| - |uc(scm(T_i, T_j), T_i)|\big)\Big).$$

**Unique-Clade-Rate scoring:** maximizing the number of preserved unique clades:

$$\frac{|uc(T_i, T_j)| + |uc(T_j, T_i)|}{|uc(scm(T_i, T_j), T_i)| + |uc(scm(T_i, T_j), T_j)|}.$$

**Collision scoring:** minimizing the number of collisions:

$$score(T_i, T_j) = -(\text{number of edges in SCM}(T_i, T_j) \text{ where a collision occured}).$$

**Unique Taxa scoring (*Swenson et al.*, *2012*):** minimizing the number of unique taxa:

$$score(T_i, T_j) = -|\mathcal{L}(T_i)\Delta\mathcal{L}(T_j)|.$$

**Overlap scoring (*Swenson et al.*, *2012*):** maximizing the number of common taxa:

$$score(T_i, T_j) = |\mathcal{L}(T_1)\cap\mathcal{L}(T_2)|.$$

**Resolution scoring (*Roshan et al.*, *2003*):** maximizing the resolution of the SCM tree:

$$score(T_i, T_j) = \frac{|\mathcal{V}(\text{SCM}(T_i, T_j))| - |\mathcal{L}(\text{SCM}(T_i, T_j))|}{|\mathcal{L}(\text{SCM}(T_i, T_j))| - 1}.$$

## Combining multiple scorings

In general, supertrees created with the GSCM using different scoring functions contain different clades. To collect as many reliable clades as possible, we compute several GSCM supertrees using different scoring functions and combine them afterwards.

Reliable clades of all possible GSCM supertrees for a given set of source trees are pairwise compatible. In contrast, bogus clades can be incompatible among different GSCM supertrees (see Fig. 2). Thus, every conflicting clade has to be a bogus clade. By removing incompatible clades we only eliminate bogus clades but none of the reliable clades from our final supertree.

Eliminating bogus clades while assembling reliable clades is done using a semi-strict consensus algorithm (*Bryant*, *2003*). It should be noted that bogus clades are only eliminated

if they induce a conflict between at least two supertrees (see Fig. 2). Hence, there is no guarantee to eliminate all bogus clades.

**Combined scoring:** Let *Combined-3* be the combination of the Collision, Unique-Clade-Rate and Unique-Clades-Lost scoring functions. Furthermore *Combined-5* combines the Collision, Unique-Clade-Rate, Unique-Clades-Lost, Overlap and Unique-Taxa scoring functions.

### Randomized GSCM

Generating many different GSCM supertrees increases the probability of both detecting all reliable clades and eliminating all bogus clades. To generate a larger number of GSCM supertrees, randomizing the tree merging order of the GSCM algorithm may be more suitable than using a variety of different tree selection scorings. To this end, we replace picking an optimal pair of trees (see Algorithm 2) by picking a random pair of trees (see Algorithm 3).

---

**Algorithm 3. Function for randomization step of the GSCM**

---

1: **function** PICKRANDOMTREEPAIR(trees $\mathscr{S} \subseteq \{T_1, T_2, \dots, T_k\}$)
2:     Randomly pick a pair of trees $\{T_i, T_j\} \subseteq \mathscr{S}$ with probability
$$P(T_i, T_j) = \frac{score(T_i, T_j)}{\sum\limits_{T_a, T_b \in \mathscr{S}, a \neq b} score(T_a, T_b)}, i \neq j$$

3:     **return** $T_i, T_j$
4: **end function**

---

Running the randomized GSCM for different scoring functions multiple ($k$) times allows us to generate a large number of supertrees containing different clades. The resulting trees are combined using a semi-strict consensus as described in the previous section. For combined scorings (Combined-$n$) with $n$ different scoring functions we calculate $\frac{k}{n}$ supertrees for each of the scoring functions and combine all $k$ supertrees using the semi-strict consensus.

## EXPERIMENTAL SETUP

### Dataset

To evaluate the different modifications of the GSCM algorithm we simulate a rooted dataset which is based on the SMIDGen protocol (*Swenson et al.*, *2010*) called *SMIDGenOG*.

We generate 30 model trees with 1,000 (500/100) taxa. For each model tree, we generate a set of 30 (15/5) clade-based source trees and four scaffold source trees containing 20%, 50%, 75%, or 100% of the taxa in the model tree (the *scaffold density*). We set up four different source tree sets: each of them containing all clade-based trees and one of the scaffold trees, respectively.

The SMIDGen protocol follows data collection processes used by systematists when gathering empirical data, e.g., the creation of several densely-sampled *clade-based source trees*, and a sparsely-sampled *scaffold source tree*. All source trees are rooted using an outgroup. Unless indicated otherwise, we strictly follow the protocol of *Swenson et al.* (*2010*), see there for more details:

1. **Generate model trees.** We generate model trees using r8s (*Sanderson*, *2003*) as described by *Swenson et al.* (*2010*). To each model tree, we add an outgroup. The branch to the outgroup gets the length of the longest path in the tree, plus a random value between 0 and 1. This outgroup placement guarantees that there exists an outgroup for every possible subtree of the model tree.

2. **Generate sequences.** Universal genes appear at the root of the model tree and do not go extinct. We simulate five universal genes along the model tree. Universal genes are used to infer scaffold trees. To simulate non-universal genes, we use a gene "birth–death" process (as described by *Swenson et al.* (*2010*)) to determine 200 subtrees (one for each gene) within the model tree for which a gene will be simulated. For comparison, the *SMIDGen* dataset evolves 100 non-universal genes. Simulating a higher number of genes increases the probability to find a valuable outgroup. Genes (both universal and non-universal) are simulated under a GTR + Gamma + Invariable Sites process along the respective tree, using Seq-Gen (*Rambaut & Grassly*, *1997*).

3. **Generate source alignments.** To generate a clade-based source alignment, we select a clade of interest from the model tree using a "birth" node selection process (as described by *Swenson et al.* (*2010*)). For each clade of interest, we select the three non-universal gene sequences with the highest taxa coverage to build the alignment. For each source alignment, we search in the model tree for an outgroup where all three non-universal genes are present and add it to the alignment.
   To generate a scaffold source alignment, we randomly select a subset of taxa from the model tree with a fixed probability (scaffold factor) and use the universal gene sequences.

4. **Estimation of source trees.** We estimate Maximum Likelihood (ML) source trees using RAxML with GTR-GAMMA default settings and 100 bootstrap replicates. We root all source trees using the outgroup, and remove the outgroups afterwards.

## Evaluation

To evaluate the accuracy of tree reconstruction methods on simulated data, a widespread method is calculating the rates of false negative (*FN*) clades and false positive (*FP*) clades between an estimated tree (supertree) and the corresponding model tree. *FN* clades are in the model tree but not in the supertree. *FP* clades are in the supertree but not in the model tree.

*FN*-rates and *FP*-rates contain information on the resolution of the supertree. Model trees are fully resolved. If it happens that the supertree is fully resolved too, we get *FN*-rate = *FP*-rate. Otherwise, if *FN*-rate > *FP*-rate the supertree is not fully resolved. Clades in the supertree that are not *FP*s are true positive (*TP*) clades.

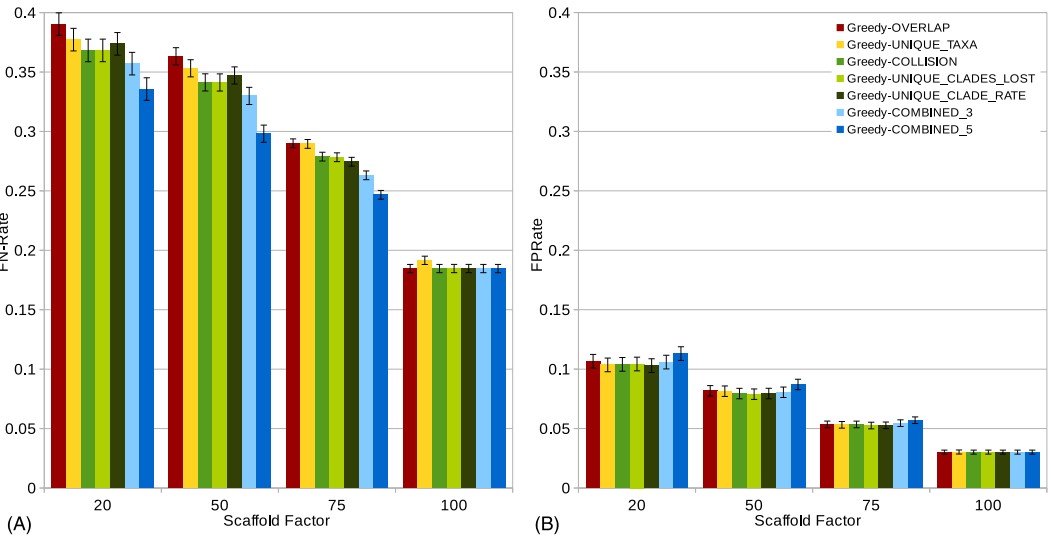

**Figure 3** *FN*-rates (A) and *FP*-rates (B) of single scorings functions (Overlap, Unique-Taxa, Collision, Unique-Clades-Lost, Unique-Clade-Rate) and their combinations (Combined-3,Combined-5) for all scaffold factors (20%, 50%, 75%, 100%) of the 1,000-taxon dataset. The Combined scorings are the semi-strict consensus of the supertrees calculated by the respective scoring functions. The error bars show the standard error.

As mentioned above, we try to improve the GSCM as a preprocessing method and thus want to maximize the number of *TP*, while keeping the number of *FP* minimal. This is reflected in the $F_1$-score:

$$F_1 = \frac{2TP}{2TP + FP + FN}.$$

We measure the statistical significance of differences between the averaged $F_1$-scores by the Wilcoxon signed-rank test with $\alpha = 0.05$. We calculate the pairwise $p$-values for all 16 scoring functions (including combined scorings and randomized scorings with 400 iterations). This leads to $\frac{16^2 - 16}{2} = 120$ significance tests. Respecting the multiple testing problem we can accept $p$-values below $\frac{0.005}{120} \approx 0.0004$ (Bonferroni correction). The complete tables can be found in Tables S1, S3 and S5.

Furthermore, Tables S2, S4 and S6 contain the number of times that each scoring function outperforms each other scoring function. Ties are reported as well.

## RESULTS AND DISCUSSION

We find the influence of scoring functions and randomization to increase with the size of the input data (as expected for greedy algorithms). Thus, in the further evaluation we only consider the larger (1,000 taxa) dataset. However, the overall effects are similar for all datasets. For the results of the 500 and 100 taxa datasets, we refer to Figs. S1–S16.

The scaffold factor highly influences the quality of the supertrees (see Figs. 3 and 4). In general, all scoring functions profit from a large scaffold tree. In particular, for a scaffold factor of 100% nearly all scorings perform equally well and better than for all other scaffold factors. A source tree that already contains all taxa simplifies the supertree computation

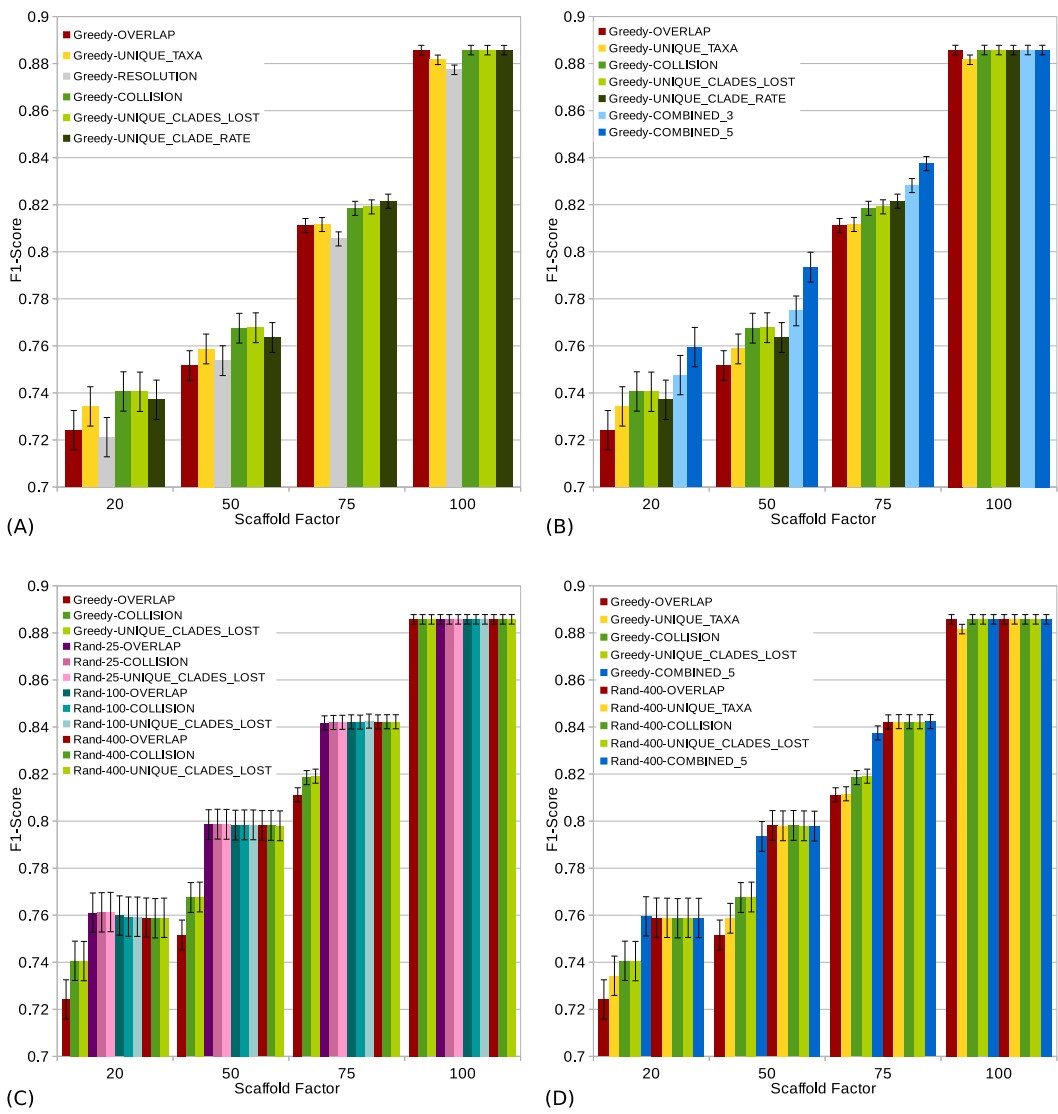

**Figure 4** $F_1$-scores (a high score is good) of different scoring functions (including combined scorings) with and without randomization for all scaffold factors (20%, 50%, 75%, 100%) of the 1,000 taxa dataset. The Combined scorings are the semi-strict consensus of the supertrees calculated by the respective scorings. The integer value after the keyword "Rand" represents the number of randomized iterations. The error bars show the standard error. (A) Comparison of single scoring functions (Overlap, Unique-Taxa, Resolution, Collision, Unique-Clades-Lost, Unique-Clades-Rate). (B) Comparison of single scoring functions (Overlap, Unique-Taxa, Collision, Unique-Clades-Lost, Unique-Clade-Rate) and their combinations (Combined-3,Combined-5). (C) Comparison of different scoring functions (Overlap, Collision, Unique-Clades-Lost) with (25, 100 and 400 random iterations) and without randomization. (D) Comparison of single (Overlap, Unique-Taxa, Collision, Unique-Clades-Lost) and combined (Combined-5) scorings. Both with 400 random iterations and without randomization.

for the GSCM algorithm. Starting with the scaffold tree and merging the remaining source trees in arbitrary order leads to the optimal solution. No collision can occur, when the taxon set of one tree is a subset of the taxon set of the other tree. However, the Resolution and Unique-Taxa scoring functions do not necessarily pick the scaffold tree in the first step and therefore do not necessarily lead to an optimal solution. In contrast, the Overlap

scoring—which does not perform well for small scaffold tree sizes (20%, 50%)—produces optimal solutions for a scaffold factor of 100%.

Comparing the different scoring functions, we find that in general, the $FN$-rate varies more than the $FP$-rate (see Fig. 3). Our presented scoring functions (Collision, Unique-Clade-Lost, Unique-Clade-Rate) decrease the $FN$-rate, without increasing the $FP$-rate (see Fig. 3). This leads to the highest $F_1$-scores for all scaffold factors (see Fig. 4A). They clearly outperform the Resolution, Overlap and Unique-Taxa scorings for scaffold factors 50% and 75%. The differences in the $F_1$-scores are significant ($p$-values below 0.000033). For a scaffold factor of 20% the improvements of our scoring functions in comparison to Unique-Taxa are not significant. For a scaffold factor of 100% the Overlap scoring function is on par with our scoring functions (all of them will return the optimal solution). The differences between Collision, Unique-Clade-Lost and Unique-Clade-Rate are not significant. Nevertheless Unique-Clade-Lost provides the most robust and input independent results. For scaffold factors of 20% and 50%, Resolution and Overlap show significantly worse ($p$-values $\leq$ 0.000006) $F_1$-scores than all other scoring functions (see Fig. 4A). There is no significant difference ($p$-values > 0.09) between Resolution and Overlap scoring. For scaffold factors of 75% and 100%, the Resolution scoring function performs significantly worse than all others. For a scaffold factor of 75%, there is no significant difference between Unique-Taxa and Overlap scoring. For a scaffold factor of 100%, the Overlap scoring function performs better than Unique-Taxa, which is still significantly better than Resolution.

Even for equally-performing scoring functions, the resulting trees are often different (except for scaffold factor 100%). Thus, we combine the GSCM supertrees computed with different scorings using the semi-strict consensus. Since the Resolution scoring function performs badly, we only combine the remaining five scoring functions. The combination of different scoring functions strongly improves the $FN$-rate. Thus, the combined supertrees have improved $F_1$-scores for all scaffold densities (see Fig. 4B). The combination of Collision, Unique-Clade-Lost, Unique-Clade-Rate, Overlap and Unique-Taxa (Combined-5) results in the best $F_1$-score. However, Combined-5 has a significantly worse $FP$-rate than all other scorings. In contrast, the combination of Collision, Unique-Clade-Lost, Unique-Clade-Rate scoring (Combined-3) shows no significant decline of the $FP$-rate.

To collect as many $TP$ clades as possible, we use a randomized tree merging order generating multiple ($k$) supertrees which are combined using the semi-strict consensus. Generally we found that randomization further improves the $F_1$-score in comparison to the single scoring functions (see Fig. 4D). Compared to the Combined-5 scoring there is only an improvement of the $F_1$-score for scaffold factors of 50% and 75%. Again, these improvements come with a significant increase of the $FP$-rate.

Already for 25 random iterations, all presented scoring functions perform on almost the same level (see Fig. 4C). As the number of random iterations increases, the difference between the reported scoring functions vanishes.

## CONCLUSION

We found that collisions not only destroy source tree clades but also introduce bogus clades to the supertree. Thus, the scoring functions that minimize the number of collisions

perform best. Combining multiple GSCM supertrees using a semi-strict consensus method helps to better resolve the supertree.

We presented three novel scoring functions (Collision, Unique-Clades-Lost, Unique-Clade-Rate) that increase the number of true positive clades and decrease the number of false positive clades of the resulting supertree. Unique-Clades-Lost score is the overall best-performing scoring function.

Combining the supertrees calculated by these three scorings using a semi-strict consensus algorithm further increases the number of true positive clades without a significant increase of the false positives.

For almost all presented scoring functions, the highest $F_1$-scores and best resolved trees are achieved using randomized GSCM. Randomization indeed increases the number of true positive clades but also significantly increases false positive clades. Thinking of GSCM as a preprocessing method, those false positive clades will have a strongly negative influence on the quality of the final supertree.

Depending on the application, "best performance" is characterized differently. The most conservative approach is our Unique-Clade-Lost scoring function which increases the $TP$-rate by 5% while decreasing the $FP$-rate by 2% compared to Overlap. To use GSCM as a preprocessing method, we recommend a combination of Collision, Unique-Clade-Lost and Unique-Clade-Rate (Combined-3) scoring. In comparison to the Overlap scoring function, this increases the number of true positive clades by 9% without a significant increase of false positive clades. The overall best ratio of true positive and false positive clades can be achieved with a combination of randomized Collision, Unique-Clade-Lost, Unique-Clade-Rate, Overlap and Unique-Taxa (Combined-5) scoring.

All presented methods are part of our platform-independent GSCM command line tool (https://bio.informatik.uni-jena.de/software/gscm/).

### Funding
Markus Fleischauer is supported by Deutsche Forschungsgemeinschaft, project BO 1910/12. The funders had no role in study design, data collection and analysis, decision to publish, or preparation of the manuscript.

### Grant Disclosures
The following grant information was disclosed by the authors:
Deutsche Forschungsgemeinschaft: BO 1910/12.

### Competing Interests
The authors declare there are no competing interests.

### Author Contributions
- Markus Fleischauer conceived and designed the experiments, performed the experiments, analyzed the data, wrote the paper, prepared figures and/or tables, performed the computation work, reviewed drafts of the paper, developed the method.

- Sebastian Böcker conceived and designed the experiments, reviewed drafts of the paper, developed the method.

## Data Availability

Lehrstuhl Bioinformatik Jena: http://bio.informatik.uni-jena.de/data/.

## Supplemental Information

Supplemental information for this article can be found online at http://dx.doi.org/10.7717/peerj.2172#supplemental-information.

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
