# Peer review of "Collecting reliable clades using the Greedy Strict Consensus Merger"

_PeerJ, doi:10.7717/peerj.2172_

## Round 0.1 · original submission · Major Revisions

This is a nice article focusing on the improvement of SuperFine to produce better results. The reviewers have suggested some changes for improvement of this manuscript. Please provide us with a revised manuscript.

Reviewer 1 ·

Basic reporting

Acceptable but there are writing problems that should be fixed.

Experimental design

Acceptable but there are writing problems that should be fixed.

Validity of the findings

Acceptable but note caveats below

Additional comments

This is a nice paper with some potentially very useful observations about how to improve SuperFine to produce better results. I have a small number of comments about the writing, most of which are probably very easily addressed in a minor revision. However, I would like to see the revision before it is accepted.

• Perhaps the most important thing is just to clarify what the authors are doing. The authors are examining the impact of how the source trees are merged within the SuperFine pipeline. But the first step of SuperFine is computing the Strict Consensus Merger (SCM) tree, and the second step refines the SCM tree using the preferred supertree method. It seems the authors are only looking at the accuracy of the SCM tree, and not of the final tree returned after the second step. If that’s so, then the importance of improving the accuracy of the first tree (the SCM) is not quite as big as if it impacted the accuracy of the entire two-step process. However, even if they are only examining the impact on the SCM tree, the study is still valuable – for two reasons. First, it might lead to improved accuracy in the second step, but even if it doesn’t, it might reduce the running time overall (because of how the second step is performed). It would help if the authors clearly discussed this issue.

• The manuscript includes text about “random clades” as being clades that are not “supported by any source tree” (see, for example, page 1, last line). Yet this is not precisely defined. Furthermore, Figure 2, which is supposed to present an example of such a clade, actually presents a bipartition. Are the halves of the bipartition are supposed to be the “random clades’? Is the definition of a clade X being “supported” by a source tree T that T contains a bipartition (S-X|X) (where S is the full set of taxa)? This would require that the source tree have all the taxa, which is too stringent a criterion. Or that T contain a bipartition (X|Y), even if Y is not identical to S-X? Even this is unreasonably stringent. For example, consider a set of source trees on a pectinate tree, with leaves labelled 1, 2,3, …, n appearing in that order. The source trees could be the true species tree on 1…30, 11…40, 21…50, 31…60, 41…70, etc., up to 71…100. There is exactly one supertree consistent with all these trees (and it is the true pectinate tree on 1…n). Yet 1…50 is true clade that would not be supported by any source tree. I’m really not sure what the definition is supposed to be. All the authors need to do is provide a rigorous definition that is understandable and explain why it’s important. It would be good if they also addressed examples like the one above and make sure that these true species tree clades are not considered “random clades”, since these are removed by their algorithm.

• The authors depend on the semi strict consensus discussed in Bryant 2003. They should define this – I’m not sure what it is, and some other readers might also not know it.

The following are the minor writing issues.
• Page 3, Line 10 in the description of Strict Consensus Merger algorithm, change “without violating any of the bipartition in” to “…any of the bipartitions in” (i.e., change “bipartition” to “bipartitions”)
• Page 5, second paragraph within section “Combining multiple scorings”;. Remove comma after “Both” and change “compatible to” to “compatible with”
• Page 6, two lines above “Results and Discussion”: remove the comma after “both”
• Page 9, three lines above “Conclusion”, the authors have “SuperFine Swenson et al. (2012)” I think they need some parentheses?

·

Basic reporting

1. No information is given on the significance of the differences of the average scores presented. The authors should add error-bars (or provide standard-error information) on the graphs/data presented, so that readers can understand the potential significance of the improvements shown. Not only should the graphs be updated to reflect the variation of error rates for a single supertree method for a single model condition, but the Results and Discussion should also be modified as necessary to convey which differences are noteworthy and which are not.

2. No information is given on how the false-positive and false-negative rates are normalized. Given that the motivation for normalization is to adjust for the effect of scaffold factor, it seems that for a given scaffold factor, the error rate (FNR or FNP) of each supertree is divided by the score of the supertree with the highest error rate for that scaffold factor. (It is not at all clear whether this normalization is done before or after the mean error rate is computed over the 30 model trees/datasets for that model condition.) Furthermore, the authors comment that the error rates are normalized "to be between zero and one." However, the FNR and FPR are already values between zero and one, so this comment seems unnecessary if not misleading. For these error measures, some sort of information on the variation of scores should be given (possibly in the form of standard-error-bars).

3. There is insufficient information given in how the source trees were generated.
a. Did the authors use the source tree datasets from Swenson et. al. 2010? Or did they generate their own sequence datasets following the SMIDGen protocol and then generate their own source trees?
b. The SMIDGen protocol only describes methods for producing UNROOTED source trees. Choosing an appropriate outgroup is a difficult process for systematists, and was not included in the SMIDGen study/protocol. Therefore, the input data considered here are likely much more accurate simply due to the fact that a "true" outgroup was created. The authors should at a minimum acknowledge this in the experimental set-up. And ideally, they might also address how they expect their results to extend to the unrooted scenario.

Experimental design

no comments

Validity of the findings

no comments

Additional comments

Beyond the major issues, I would like to bring minor issues / typos to your attention:

Grammatical edit (Introduction, paragraph 1, last sentence): move "rather" later in the sentence so that it reads '...the "raw" sequence data rather than the trees...".

Grammatical edit (Introduction, last paragraph, second sentence): add "rather" before "... than using the same number..."

Grammatical edit (Methods, preliminaries paragraph, third sentence): add "a" so that it reads "...called a taxon."

Grammatical edit (Methods, preliminaries paragraph, fifth sentence): add "an" so that it reads "...an inner vertex."

Suggested clarification (Methods, preliminaries paragraph, 6th sentence): add text so that it is clear that the clade is defined by the leafs whose path to the root node must pass through the node inducing the clade.

Typo (Methods, preliminaries paragraph, second to last sentence): replace "is" with "as".

Suggested clarification (Methods, preliminaries paragraph): When describing the induced subtrees, it is clearer to speak only of edge contractions, and avoid vertex deletions. (As vertex deletions leave it unclear what should happen with any edges that have that vertex as an endpoint. Alternatively, you could specify that the adjacent edges are replaced with single edge.)

Grammatical edit (Methods, SCM first paragraph, first sentence): replace "set" with "sets".
Grammatical edit (Methods, combining multiple scorings first paragraph, first sentence): replace "much" with "many".

Suggested clarification (Methods, combining multiple scorings second paragraph, third sentence): change "causes" to "can lead to".

Grammatical edit (Methods, combining multiple scorings second paragraph, 5th sentence): change "compatible to all clades" to "compatible with all clades".

Stylistic suggestion (Methods, randomized GSCM first paragraph, second sentence): change "coming up with lots of different" to "using a variety of".

Suggested clarification (Experimental Setup, 3rd paragraph, second sentence): Change "but should be" to "but are in the model tree".

Grammatical edit (Results and Discussion, second paragraph, 4th sentence): change " therefore not necessarily leads to" to "therefore does not necessarily lead to".

---

## Round 0.2 · Minor Revisions

Thanks for resubmitting this manuscript. One of the reviewers has raised concerns about the writing - please go through your manuscript and improve whatever you can. This will improve your manuscript's readability.

Reviewer 1 ·

Basic reporting

No comments (acceptable)

Experimental design

No comments (acceptable)

Validity of the findings

No comments (acceptable)

Additional comments

Very good revision and you have addressed all my concerns and questions. However there are still problems with the writing. I do not wish to provide a full list, but here are two examples that appear on page 4:

page 4: first paragraph, last sentence. You have written "In case of a collision...the colliding subtrees are inserted as polytomy"

page 4, second paragraph, first sentence. You have written "If collisions of different merging steps...the polytomy created by the first collision may causes"

Other than the writing (which are minor problems but should still be fixed), I have no other concerns.

·

Basic reporting

No Comments

Experimental design

No Comments

Validity of the findings

No Comments

---

## Round 0.3 · accepted · Accept

Congratulations! After appropriate changes this manuscript is now accepted.